# Blind Regression: Nonparametric Regression for Latent Variable Models via Collaborative Filtering

**Christina E. Lee**    **Yihua Li**    **Devavrat Shah**    **Dogyoon Song**
Laboratory for Information and Decision Systems
Department of Electrical Engineering and Computer Science
Massachusetts Institute of Technology
{celee, liyihua, devavrat, dgsong}@mit.edu

## Abstract

We introduce the framework of *blind regression* motivated by *matrix completion* for recommendation systems: given $m$ users, $n$ movies, and a subset of user-movie ratings, the goal is to predict the unobserved user-movie ratings given the data, i.e., to complete the partially observed matrix. Following the framework of non-parametric statistics, we posit that user $u$ and movie $i$ have features $x_1(u)$ and $x_2(i)$ respectively, and their corresponding rating $y(u, i)$ is a noisy measurement of $f(x_1(u), x_2(i))$ for some unknown function $f$. In contrast with classical regression, the features $x = (x_1(u), x_2(i))$ are not observed, making it challenging to apply standard regression methods to predict the unobserved ratings.

Inspired by the classical Taylor's expansion for differentiable functions, we provide a prediction algorithm that is consistent for all Lipschitz functions. In fact, the analysis through our framework naturally leads to a variant of collaborative filtering, shedding insight into the widespread success of collaborative filtering in practice. Assuming each entry is sampled independently with probability at least $\max(m^{-1+\delta}, n^{-1/2+\delta})$ with $\delta > 0$, we prove that the expected fraction of our estimates with error greater than $\epsilon$ is less than $\gamma^2/\epsilon^2$ plus a polynomially decaying term, where $\gamma^2$ is the variance of the additive entry-wise noise term.

Experiments with the MovieLens and Netflix datasets suggest that our algorithm provides principled improvements over basic collaborative filtering and is competitive with matrix factorization methods.

## 1   Introduction

In this paper, we provide a statistical framework for performing nonparametric regression over latent variable models. We are initially motivated by the problem of matrix completion arising in the context of designing recommendation systems. In the popularized setting of Netflix, there are $m$ users, indexed by $u \in [m]$, and $n$ movies, indexed by $i \in [n]$. Each user $u$ has a rating for each movie $i$, denoted as $y(u, i)$. The system observes ratings for only a small fraction of user-movie pairs. The goal is to predict ratings for the rest of the unknown user-movie pairs, i.e., to complete the partially observed $m \times n$ rating matrix. To be able to obtain meaningful predictions from the partially observed matrix, it is essential to impose a structure on the data.

We assume each user $u$ and movie $i$ is associated to features $x_1(u) \in \mathcal{X}_1$ and $x_2(i) \in \mathcal{X}_2$ for some compact metric spaces $\mathcal{X}_1, \mathcal{X}_2$ equipped with Borel probability measures. Following the philosophy of non-parametric statistics, we assume that there exists some function $f : \mathcal{X}_1 \times \mathcal{X}_2 \to \mathbb{R}$ such that the rating of user $u$ for movie $i$ is given by

$$y(u, i) = f(x_1(u), x_2(i)) + \eta_{ui}, \tag{1}$$

where $\eta_{ui}$ is some independent bounded noise. We observe ratings for a subset of the user-movie pairs, and the goal is to use the given data to predict $f(x_1(u), x_2(i))$ for all $(u, i) \in [m] \times [n]$ whose rating is unknown. In classical nonparametric regression, we observe input features $x_1(u), x_2(i)$ along with the rating $y(u, i)$ for each datapoint, and thus we can approximate the function $f$ well using local approximation techniques as long as $f$ satisfies mild regularity conditions. However, in our setting, we do not observe the latent features $x_1(u), x_2(i)$, but instead we only observe the indices $(u, i)$. Therefore, we use *blind regression* to refer to the challenge of performing regression with unobserved latent input variables. This paper addresses the question, does there exist a meaningful prediction algorithm for general nonparametric regression when the input features are unobserved?

**Related Literature.** Matrix completion has received enormous attention in the past decade. Matrix factorization based approaches, such as low-rank approximation, and neighborhood based approaches, such as collaborative filtering, have been the primary ways to address the problem. In the recent years, there has been exciting intellectual development in the context of matrix factorization based approaches. Since any matrix can be factorized, its entries can be described by a function $f$ in (1) with the form $f(x_1, x_2) = x_1^T x_2$, and the goal of factorization is to recover the latent features for each row and column. [25] was one of the earlier works to suggest the use of low-rank matrix approximation, observing that a low-rank matrix has a comparatively small number of free parameters. Subsequently, statistically efficient approaches were suggested using optimization based estimators, proving that matrix factorization can fill in the missing entries with sample complexity as low as $rn \log n$, where $r$ is the rank of the matrix [5, 23, 11, 21, 10]. There has been an exciting line of ongoing work to make the resulting algorithms faster and scalable [7, 17, 4, 15, 24, 20].

Many of these approaches are based on the structural assumption that the underlying matrix is *low-rank* and the matrix entries are reasonably "incoherent". Unfortunately, the low-rank assumption may not hold in practice. The recent work [8] makes precisely this observation, showing that a simple non-linear, monotonic transformation of a low-rank matrix could easily produce an effectively high-rank matrix, despite few free model parameters. They provide an algorithm and analysis specific to the form of their model, which achieves sample complexity of $O((mn)^{2/3})$. However, their algorithm only applies to functions $f$ which are a nonlinear monotonic transformation of the inner product of the latent features. [6] proposes the universal singular value thresholding estimator (USVT), and they provide an analysis under a similar model in which they assume $f$ to be a bounded Lipschitz function. They achieve a sample complexity, or the required fraction of measurements over the total $mn$ entries, which scales with the latent space dimension $q$ according to $\Omega\left(m^{-2/(q+2)}\right)$ for a square matrix, whereas we achieve a sample complexity of $\Omega(m^{-1/2+\delta})$ (which is independent of $q$) as long as the latent dimension scales as $o(\log n)$.

The term collaborative filtering was coined in [9], and this technique is widely used in practice due to its simplicity and ability to scale. There are two main paradigms in neighborhood-based collaborative filtering: the user-user paradigm and the item-item paradigm. To recommend items to a user in the user-user paradigm, one first looks for similar users, and then recommends items liked by those similar users. In the item-item paradigm, in contrast, items similar to those liked by the user are found and subsequently recommended. Much empirical evidence exists that the item-item paradigm performs well in many cases [16, 14, 22], however the theoretical understanding of the method has been limited. In recent works, Latent mixture models or cluster models have been introduced to explain the collaborative filtering algorithm as well as the empirically observed superior performance of item-item paradigms, c.f. [12, 13, 1, 2, 3]. However, these results assume a specific parametric model, such as a mixture distribution model for preferences across users and movies. We hope that by providing an analysis for collaborative filtering within our broader nonparametric model, we can provide a more complete understanding of the potentials and limitations of collaborative filtering.

The algorithm that we propose in this work is inspired by local functional approximations, specifically Taylor's approximation and classical kernel regression, which also relies on local smoothed approximations, c.f. [18, 26]. However, since kernel regression and other similar methods use explicit knowledge of the input features, their analysis and proof techniques do not extend to our context of Blind regression, in which the features are latent. Although our estimator takes a similar form of computing a convex combination of nearby datapoints weighted according to a function of the latent distance, the analysis required is entirely different.

**Contributions.** The key contribution of our work is in providing a statistical framework for nonparametric regression over latent variable models. We refrain from any specific modeling assumptions on $f$, keeping mild regularity conditions aligned with the philosophy of non-parametric statistics. We assume that the latent features are drawn independently from an identical distribution (IID) over bounded metric spaces; the function $f$ is Lipschitz with respect to the latent spaces; entries are observed independently with some probability $p$; and the additive noise in observations is independently distributed with zero mean and bounded support. In spite of the minimal assumptions of our model, we provide a consistent matrix completion algorithm with finite sample error bounds. Furthermore, as a coincidental by-product, we find that our framework provides an explanation of the practical mystery of "why collaborative filtering algorithms work well in practice".

There are two conceptual parts to our algorithm. First, we derive an estimate of $f(x_1(u), x_2(i))$ for an unobserved index pair $(u, i)$ by using first order local Taylor approximation expanded around the points corresponding to $(u, i')$, $(u', i)$, and $(u', i')$. This leads to estimation that

$$\hat{y}(u, i) \equiv y(u', i) + y(u, i') - y(u', i') \approx f(x_1(u), x_2(i)), \qquad (2)$$

as long as $x_1(u')$ is close to $x_1(u)$ or $x_2(i')$ is close to $x_2(i)$. In kernel regression, distances between input features are used to upper bound the error of individual estimates, but since the latent features are not observed, we need another method to determine which of these estimates are reliable.

Secondly, under mild regularity conditions, we upper bound the squared error of the estimate in (2) by the the variance of the squared difference between commonly observed entries in rows $(u, v)$ or columns $(i, j)$. We empirically estimate this quantity and use it similarly to distance in the latent space in order to appropriately weight individual estimates to a final prediction. If we choose only the datapoints with minimum empirical row variance, we recover user-user nearest neighbor collaborative filtering. Inspired by kernel regression, we also propose using computing the weights according to a Gaussian kernel applied to the minimum of the row or column sample variances.

As the main technical result, we show that the user-user nearest neighbor variant of collaborative filtering method with our similarity metric yields a consistent estimator for any Lipschitz function as long as we observe $\max(m^{-1+\delta}, n^{-1/2+\delta})$ fraction of the matrix with $\delta > 0$. In the process, we obtain finite sample error bounds, whose details are stated in Theorem 1. We compared the Gaussian kernel variant of our algorithm to classic collaborative filtering algorithms and a matrix factorization based approach (softImpute) on predicting user-movie ratings for the Netflix and MovieLens datasets. Experiments suggest that our method improves over existing collaborative filtering methods, and sometimes outperforms matrix-factorization-based approaches depending on the dataset.

## 2 Setup

**Operating assumptions.** There are $m$ users and $n$ movies. The rating of user $u \in [m]$ for movie $i \in [n]$ is given by (1), taking the form $y(u, i) = f(x_1(u), x_2(i)) + \eta_{u,i}$. We make the following assumptions.

(a) $\mathcal{X}_1$ and $\mathcal{X}_2$ are compact metric spaces endowed with metric $d_{\mathcal{X}_1}$ and $d_{\mathcal{X}_2}$ respectively:

$$d_{\mathcal{X}_1}(x_1, x_1') \leq B_{\mathcal{X}}, \ \forall \ x_1, x_1' \in \mathcal{X}_1, \text{ and } d_{\mathcal{X}_2}(x_2, x_2') \leq B_{\mathcal{X}}, \ \forall \ x_2, x_2' \in \mathcal{X}_2. \quad (3)$$

(b) $f : \mathcal{X}_1 \times \mathcal{X}_2 \to \mathbb{R}$ is $L-$Lipschitz with respect to $\infty$-product metric:

$$|f(x_1, x_2) - f(x_1', x_2')| \leq L \max\{d_{\mathcal{X}_1}(x_1, x_1'), d_{\mathcal{X}_2}(x_2, x_2')\}, \ \forall x_1, x_1' \in \mathcal{X}_1, x_2, x_2' \in \mathcal{X}_2.$$

(c) The latent features of each user $u$ and movie $i$, $x_1(u)$ and $x_2(i)$, are sampled independently according to Borel probability measures $P_{\mathcal{X}_1}$ and $P_{\mathcal{X}_2}$ on $(\mathcal{X}_1, T_{\mathcal{X}_1})$ and $(\mathcal{X}_2, T_{\mathcal{X}_2})$, where $T_{\mathcal{X}}$ denotes the Borel $\sigma$-algebra of a metric space $\mathcal{X}$.

(d) The additive noise for all data points are independent and bounded with mean zero and variance $\gamma^2$: for all $u \in [m]$, $i \in [n]$,

$$\eta_{u,i} \in [-B_\eta, B_\eta], \qquad \mathbb{E}[\eta_{u,i}] = 0, \qquad \text{Var}[\eta_{u,i}] = \gamma^2. \qquad (4)$$

(e) Rating of each entry is revealed (observed) with probability $p$, independently.

**Notation.** Let random variable $M_{ui} = 1$ if the rating of user $u$ and movie $i$ is revealed and $0$ otherwise. $M_{ui}$ is an independent Bernoulli random variable with parameter $p$. Let $N_1(u)$ denote the set of column indices of observed entries in row $u$. Similarly, let $N_2(i)$ denote the set of row indices of observed entries in column $i$. That is,

$$N_1(u) \triangleq \{i : M(u,i) = 1\} \text{ and } N_2(i) \triangleq \{u : M(u,i) = 1\}. \tag{5}$$

For rows $v \neq u$, $N_1(u,v) \triangleq N_1(u) \cap N_1(v)$ denotes column indices of commonly observed entries of rows $(u,v)$. For columns $i \neq j$, $N_2(i,j) \triangleq N_2(i) \cap N_2(j)$ denotes row indices of commonly observed entries of columns $(i,j)$. We refer to this as the *overlap* between two rows or columns.

## 3  Algorithm Intuition

**Local Taylor Approximation.** We propose a prediction algorithm for unknown ratings based on insights from the classical Taylor approximation of a function. Suppose $\mathcal{X}_1 \cong \mathcal{X}_2 \cong \mathbb{R}$, and we wish to predict unknown rating, $f(x_1(u), x_2(i))$, of user $u \in [m]$ for movie $i \in [n]$. Using the first order Taylor expansion of $f$ around $(x_1(v), x_2(j))$ for some $u \neq v \in [m], i \neq j \in [n]$, it follows that

$$f(x_1(u), x_2(i)) \approx f(x_1(v), x_2(j)) + (x_1(u) - x_1(v))\frac{\partial f(x_1(v), x_2(j))}{\partial x_1} + (x_2(i) - x_2(j))\frac{\partial f(x_1(v), x_2(j))}{\partial x_2}.$$

We are not able to directly compute this expression, as we do not know the latent features, the function $f$, or the partial derivatives of $f$. However, we can again apply Taylor expansion for $f(x_1(v), x_2(i))$ and $f(x_1(u), x_2(j))$ around $(x_1(v), x_2(j))$, which results in a set of equations with the same unknown terms. It follows from rearranging terms and substitution that

$$f(x_1(u), x_2(i)) \approx f(x_1(v), x_2(i)) + f(x_1(u), x_2(j)) - f(x_1(v), x_2(j)),$$

as long as the first order Taylor approximation is accurate. Thus if the noise term in (1) is small, we can approximate $f(x_1(u), x_2(i))$ by using observed ratings $y(v,j), y(u,j)$ and $y(v,i)$ according to

$$\hat{y}(u,i) = y(u,j) + y(v,i) - y(v,j). \tag{6}$$

**Reliability of Local Estimates.** We will show that the variance of the difference between two rows or columns upper bounds the estimation error. Therefore, in order to ensure the accuracy of the above estimate, we use empirical observations to estimate the variance of the difference between two rows or columns, which directly relates to an error bound. By expanding (6) according to (1), the error $f(x_1(u), x_2(i)) - \hat{y}(u,i)$ is equal to

$$(f(x_1(u), x_2(i)) - f(x_1(v), x_2(i))) - (f(x_1(u), x_2(j)) - f(x_1(v), x_2(j))) - \eta_{vi} + \eta_{vj} - \eta_{uj}.$$

If we condition on $x_1(u)$ and $x_1(v)$,

$$\mathbb{E}\left[(\text{Error})^2 \mid x_1(u), x_1(v)\right] = 2\,\text{Var}_{x \sim X_2}\left[f(x_1(u), x) - f(x_1(v), x) \mid x_1(u), x_1(v)\right] + 3\gamma^2.$$

Similarly, if we condition on $x_2(i)$ and $x_2(j)$ it follows that the expected squared error is bounded by the variance of the difference between the ratings of columns $i$ and $j$. This theoretically motivates weighting the estimates according to the variance of the difference between the rows or columns.

## 4  Algorithm Description

We provide the algorithm for predicting an unknown entry in position $(u,i)$ using available data. Given a parameter $\beta \geq 2$, define $\beta$-*overlapping* neighbors of $u$ and $i$ respectively as

$$\mathcal{S}_u^\beta(i) = \{v \text{ s.t. } v \in N_2(i),\ v \neq u,\ |N_1(u,v)| \geq \beta\},$$
$$\mathcal{S}_i^\beta(u) = \{j \text{ s.t. } j \in N_1(u),\ j \neq i,\ |N_2(i,j)| \geq \beta\}.$$

For each $v \in \mathcal{S}_u^\beta(i)$, compute the empirical row variance between $u$ and $v$,

$$s_{uv}^2 = \frac{1}{2|N_1(u,v)|(|N_1(u,v)| - 1)} \sum_{i,j \in N_1(u,v)} \left((y(u,i) - y(v,i)) - (y(u,j) - y(v,j))\right)^2. \tag{7}$$

Similarly, compute empirical column variances between $i$ and $j$, for all $j \in \mathcal{S}_i^\beta(u)$,

$$s_{ij}^2 = \frac{1}{2|N_2(i,j)|(|N_2(i,j)|-1)} \sum_{u,v \in N_2(i,j)} \left( (y(u,i) - y(u,j)) - (y(v,i) - y(v,j)) \right)^2. \quad (8)$$

Let $B^\beta(u,i)$ denote the set of positions $(v,j)$ such that the entries $y(v,j)$, $y(u,j)$ and $y(v,i)$ are observed, and the commonly observed ratings between $(u,v)$ and between $(i,j)$ are at least $\beta$.

$$B^\beta(u,i) = \left\{ (v,j) \in \mathcal{S}_u^\beta(i) \times \mathcal{S}_i^\beta(u) \text{ s.t. } M(v,j) = 1 \right\}.$$

Compute the final estimate as a convex combination of estimates derived in (6) for $(v,j) \in B^\beta(u,i)$,

$$\hat{y}(u,i) = \frac{\sum_{(v,j) \in B^\beta(u,i)} w_{ui}(v,j) \left( y(u,j) + y(v,i) - y(v,j) \right)}{\sum_{(v,j) \in B^\beta(u,i)} w_{ui}(v,j)}, \quad (9)$$

where the weights $w_{ui}(v,j)$ are defined as a function of (7) and (8). We proceed to discuss a few choices for the weight function, each of which results in a different algorithm.

**User-User or Item-Item Nearest Neighbor Weights.** We can evenly distribute the weights only among entries in the nearest neighbor row, i.e., the row with minimal empirical variance,

$$w_{vj} = \mathbb{I}(v = u^*), \text{ for } u^* \in \underset{v \in \mathcal{S}_u^\beta(i)}{\arg\min} s_{uv}^2.$$

If we substitute these weights in (9), we recover an estimate which is asymptotically equivalent to the mean-adjusted variant of the classical user-user nearest neighbor (collaborative filtering) algorithm,

$$\hat{y}(u,i) = y(u^*,i) + m_{uu^*},$$

where $m_{uu^*}$ is the empirical mean of the difference of ratings between rows $u$ and $u^*$. For any $u,v$,

$$m_{uv} = \frac{1}{|N_1(u,v)|} \sum_{j \in N_1(u,v)} (y(u,j) - y(v,j)).$$

Equivalently, we can evenly distribute the weights among entries in the nearest neighbor columns, i.e., the column with minimal empirical variance, recovering the classical mean-adjusted item-item nearest neighbor collaborative filtering algorithm. Theorem 1 proves that this simple algorithm produces a consistent estimator, and we provide the finite sample error analysis. Due to the similarities, our analysis also directly implies the proof of correctness and consistency for the classic user-user and item-item collaborative filtering method.

**User-Item Gaussian Kernel Weights.** Inspired by kernel regression, we introduce a variant of the algorithm which computes the weights according to a Gaussian kernel function with bandwith parameter $\lambda$, substituting in the minimum row or column sample variance as a proxy for the distance,

$$w_{vj} = \exp(-\lambda \min\{s_{uv}^2, s_{ij}^2\}).$$

When $\lambda = \infty$, the estimate only depends on the basic estimates whose row or column has the minimum sample variance. When $\lambda = 0$, the algorithm equally averages all basic estimates. We applied this variant of our algorithm to both movie recommendation and image inpainting data, which show that our algorithm improves upon user-user and item-item classical collaborative filtering.

**Connections to Cosine Similarity Weights.** In our algorithm, we determine reliability of estimates as a function of the sample variance, which is equivalent to the squared distance of the mean-adjusted values. In classical collaborative filtering, cosine similarity is commonly used, which can be approximated as a different choice of the weight kernel over the squared difference.

## 5  Main Theorem

Let $E \subset [m] \times [n]$ denote the set of user-movie pairs for which the algorithm predicts a rating. For $\varepsilon > 0$, the overall $\varepsilon$-risk of the algorithm is the fraction of estimates whose error is larger than $\varepsilon$,

$$\text{Risk}_\varepsilon = \frac{1}{|E|} \sum_{(u,i) \in E} \mathbb{I}(|f(x_1(u), x_2(i)) - \hat{y}(u,i)| > \varepsilon). \quad (10)$$

In Theorem 1, we upper bound the expected $\varepsilon$-Risk, proving that the user-user nearest neighbor estimator is consistent, i.e., in the presence of no noise, estimates converge to the true values as $m, n$ go to infinity. We may assume $m \leq n$ without loss of generality.

**Theorem 1.** *For a fixed $\varepsilon > 0$, as long as $p \geq \max\{m^{-1+\delta}, n^{-1/2+\delta}\}$ (where $\delta > 0$), for any $\rho = \omega(n^{-2\delta/3})$, the user-user nearest-neighbor variant of our method with $\beta = np^2/2$ achieves*

$$\mathbb{E}[\text{Risk}_\varepsilon] \leq \frac{3\rho + \gamma^2}{\varepsilon^2}\left(1 + \frac{3 \cdot 2^{1/3}}{\varepsilon}n^{-\frac{2}{3}\delta}\right) + O\left(\exp\left(-\frac{1}{4}Cm^\delta\right) + m^\delta \exp\left(-\frac{1}{5B^2}n^{\frac{2}{3}\delta}\right)\right).$$

*where $B = 2(LB_{\mathcal{X}} + B_\eta)$, and $C = h\left(\sqrt{\frac{\rho}{L^2}}\right) \wedge \frac{1}{6}$ for $h(r) := \inf_{x_0 \in \mathcal{X}_1} \mathbb{P}_{\mathbf{x} \sim P_{\mathcal{X}_1}}\left(d_{\mathcal{X}_1}(\mathbf{x}, x_0) \leq r\right)$.*

For a generic $\beta$, we can also provide precise error bounds of a similar form, with modified rates of convergence. Choosing $\beta$ to grow with $np^2$ ensures that as $n$ goes to infinity, the required overlap between rows also goes to infinity, thus the empirical mean and variance computed in the algorithm converge precisely to the true mean and variance. The parameter $\rho$ in Theorem 1 is introduced purely for the purpose of analysis, and is not used within the implementation of the the algorithm.

The function $h$ behaves as a lower bound of the cumulative distribution function of $P_{\mathcal{X}_1}$, and it always exists under our assumptions that $\mathcal{X}_1$ is compact. It is used to ensure that for any $u \in [m]$, with high probability, there exists another row $v \in \mathcal{S}_u^\beta(i)$ such that $d_{\mathcal{X}_1}(x_1(u), x_1(v))$ is small, implying by the Lipschitz condition that we can use the values of row $v$ to approximate the values of row $u$ well. For example, if $P_{\mathcal{X}_1}$ is a uniform distribution over a unit cube in $q$ dimensional Euclidean space, then $h(r) = \min(1, r)^q$, and our error bound becomes meaningful for $n \geq (L^2/\rho)^{q/2\delta}$. On the other hand, if $P_{\mathcal{X}_1}$ is supported over finitely many points, then $h(r) = \min_{\mathbf{x} \in \text{supp}(P_{\mathcal{X}_1})} P_{\mathcal{X}_1}(\mathbf{x})$ is a positive constant, and the role of the latent dimension becomes irrelevant. Intuitively, the "geometry" of $P_{\mathcal{X}_1}$ through $h$ near 0 determines the impact of the latent space dimension on the sample complexity, and our results hold as long as the latent dimension $q = o(\log n)$.

## 6 Proof Sketch

For any evaluation set of unobserved entries $E$, the expectation of $\varepsilon$-risk is

$$\mathbb{E}[\text{Risk}_\varepsilon] = \frac{1}{|E|}\sum_{(u,i) \in E} \mathbb{P}(|f(\mathbf{x}_1(u), \mathbf{x}_2(i)) - \hat{y}(u, i)| > \varepsilon) = \mathbb{P}(|f(\mathbf{x}_1(u), \mathbf{x}_2(i)) - \hat{y}(u, i)| > \varepsilon),$$

because the indexing of the entries are exchangeable and identically distributed. To bound the expected risk, it is sufficient to provide a tail bound for the probability of the error. For any fixed $a, b \in \mathcal{X}_1$, and random variable $\mathbf{x} \sim P_{\mathcal{X}_2}$, we denote the mean and variance of the difference $f(a, \mathbf{x}) - f(b, \mathbf{x})$ by

$$\mu_{ab} \triangleq \mathbb{E}_{\mathbf{x}}[f(a, \mathbf{x}) - f(b, \mathbf{x})] = \mathbb{E}[m_{uv} | \mathbf{x}_1(u) = a, \mathbf{x}_1(v) = b],$$
$$\sigma_{ab}^2 \triangleq \text{Var}_{\mathbf{x}}[f(a, \mathbf{x}) - f(b, \mathbf{x})] = \mathbb{E}[s_{uv}^2 | \mathbf{x}_1(u) = a, \mathbf{x}_1(v) = b] - 2\gamma^2,$$

which we point out is also equivalent to the expectation of the empirical means and variances computed by the algorithm when we condition on the latent representations of the users. The computation of $\hat{y}(u, i)$ involves two steps: first the algorithm determines the neighboring row with the minimum sample variance, $u^* = \arg\min_{v \in \mathcal{S}_u^\beta(i)} s_{uv}^2$, and then it computes the estimate by adjusting according to the empirical mean, $\hat{y}(u, i) := y(u^*, i) + m_{uu^*}$.

The proof involves three key steps, each stated within a lemma. Lemma 1 proves that with high probability the observations are dense enough such that there is sufficient number of rows with overlap of entries larger than $\beta$, i.e., the number of the candidate rows, $|\mathcal{S}_u^\beta(i)|$, concentrates around $(m-1)p$. This relies on concentration of Binomial random variables via Chernoff's bound.

**Lemma 1.** *Given $p > 0$, $2 \leq \beta \leq np^2/2$ and $\alpha > 0$, for any $(u, i) \in [m] \times [n]$,*

$$\mathbb{P}\left(|\mathcal{S}_u^\beta(i)| \notin (1 \pm \alpha)(m-1)p\right) \leq 2\exp\left(-\frac{\alpha^2(m-1)p}{3}\right) + (m-1)\exp\left(-\frac{np^2}{8}\right).$$

Lemma 2 proves that since the latent features are sampled iid from a bounded metric space, for any index pair $(u, i)$, there exists a "good" neighboring row $v \in \mathcal{S}_u^\beta(i)$, whose $\sigma_{\mathbf{x}_1(u)\mathbf{x}_1(v)}^2$ is small.

**Lemma 2.** *Consider $u \in [n]$ and set $\mathcal{S} \subset [n] \setminus \{u\}$. Then for any $\rho > 0$,*

$$\mathbb{P}\left(\min_{v \in \mathcal{S}} \sigma^2_{\mathbf{x}_1(u)\mathbf{x}_1(v)} > \rho\right) \leq \left(1 - h\left(\sqrt{\frac{\rho}{L^2}}\right)\right)^{|\mathcal{S}|},$$

*where $h(r) := \inf_{x_0 \in \mathcal{X}_1} \mathbb{P}_{\mathbf{x} \sim P_{\mathcal{X}_1}} \left(d_{\mathcal{X}_1}(\mathbf{x}, x_0) \leq r\right)$.*

Subsequently, conditioned on the event that $|\mathcal{S}^\beta_u(i)| \approx (m-1)p$, Lemmas 3 and 4 prove that the sample mean and sample variance of the differences between two rows concentrate around the true mean and true variance with high probability. This involves using the Lipschitz and bounded assumptions on $f$ and $\mathcal{X}_1$, as well as the Bernstein and Maurer-Pontil inequalities.

**Lemma 3.** *Given $u, v \in [m]$, $i \in [n]$ and $\beta \geq 2$, for any $\alpha > 0$,*

$$\mathbb{P}\left(\left|\mu_{\mathbf{x}_1(u)\mathbf{x}_1(v)} - m_{uv}\right| > \alpha \mid v \in \mathcal{S}^\beta_u(i)\right) \leq \exp\left(-\frac{3\beta\alpha^2}{6B^2 + 2B\alpha}\right),$$

*where recall that $B = 2(LB_{\mathcal{X}} + B_\eta)$.*

**Lemma 4.** *Given $u \in [m]$, $i \in [n]$, and $\beta \geq 2$, for any $\rho > 0$,*

$$\mathbb{P}\left(\left|s^2_{uv} - (\sigma^2_{\mathbf{x}_1(u)\mathbf{x}_1(v)} + 2\gamma^2)\right| > \rho \mid v \in \mathcal{S}^\beta_u(i)\right) \leq 2\exp\left(-\frac{\beta\rho^2}{4B^2(2LB^2_{\mathcal{X}} + 4\gamma^2 + \rho)}\right),$$

*where recall that $B = 2(LB_{\mathcal{X}} + B_\eta)$.*

Given that there exists a neighbor $v \in \mathcal{S}^\beta_u(i)$ whose true variance $\sigma^2_{\mathbf{x}_1(u)\mathbf{x}_1(v)}$ is small, and conditioned on the event that all the sample variances concentrate around the true variance, it follows that the true variance between $u$ and its nearest neighbor $u^*$ is small with high probability. Finally, conditioned on the event that $|\mathcal{S}^\beta_u(i)| \approx (m-1)p$ and the true variance between the target row and the nearest neighbor row is small, we provide a bound on the tail probability of the estimation error by using Chevyshev inequalities. The only term in the error probability which does not decay to zero is the error from Chebyshev's inequality, which dominates the final expression, leading to the final result.

## 7    Experiments

We evaluated the performance of our algorithm to predict user-movie ratings on the MovieLens 1M and Netflix datasets. For the implementation of our method, we used user-item Gaussian kernel weights for the final estimator. We chose overlap parameter $\beta = 2$ to ensure the algorithm is able to compute an estimate for all missing entries. When $\beta$ is larger, the algorithm enforces rows (or columns) to have more commonly rated movies (or users). Although this increases the reliability of the estimates, it also reduces the fraction of entries for which the estimate is defined. We optimized the $\lambda$ bandwidth parameter of the Gaussian kernel by evaluating the method with multiple values for $\lambda$ and choosing the value which minimizes the error.

We compared our method with user-user collaborative filtering, item-item collaborative filtering, and softImpute from [20]. We chose the classic mean-adjusted collaborative filtering method, in which the weights are proportional to the cosine similarity of pairs of users or items (i.e. movies). SoftImpute is a matrix-factorization-based method which iteratively replaces missing elements in the matrix with those obtained from a soft-thresholded SVD.

For both MovieLens and Netflix data sets, the ratings are integers from 1 to 5. From each dataset, we generated 100 smaller user-movie rating matrices, in which we randomly subsampled 2000 users and 2000 movies. For each rating matrix, we randomly select and withhold a percentage of the known ratings for the test set, while the remaining portion of the data set is revealed to the algorithm for computing the estimates. After the algorithm computes its predictions for unrevealed movie-user pairs, we evaluate the Root Mean Squared Error (RMSE) of the predictions compared with the withheld test set, where RMSE is defined as the square root of the mean of squared prediction error over the evaluation set. Figure 1 plots the RMSE of our method along with classic collaborative filtering and softImpute evaluated against $10\%$, $30\%$, $50\%$, and $70\%$ withheld test sets. The RMSE is averaged over 100 subsampled rating matrices, and $95\%$ confidence intervals are provided.

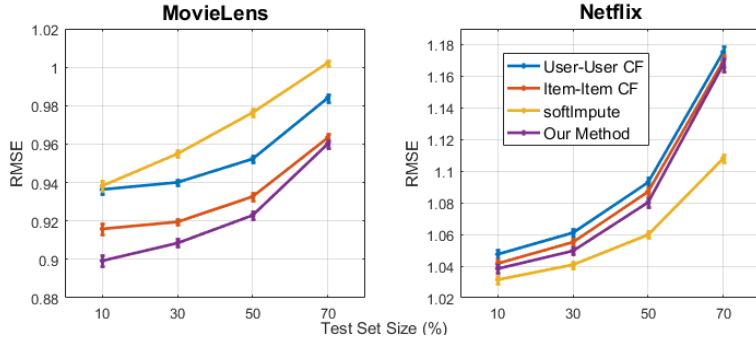

Figure 1: Performance of algorithms on Netflix and MovieLens datasets with 95% confidence interval. $\lambda$ values used by our algorithm are 2.8 (10%), 2.3 (30%), 1.7 (50%), 1 (70%) for MovieLens, and 1.8 (10%), 1.7 (30%), 1.6 (50%), 1.5 (70%) for Netflix.

Figure 1 suggests that our algorithm achieves a systematic improvement over classical user-user and item-item collaborative filtering. SoftImpute performs the worst on the MovieLens dataset, but it performs the best on the Netflix dataset. This behavior could be due to different underlying assumptions of low rank for matrix factorization methods as opposed to Lipschitz for collaborative filtering methods, which could lead to dataset dependent performance outcomes.

## 8    Discussion

We introduced a generic framework of blind regression, i.e., nonparametric regression over latent variable models. We allow the model to be any Lipschitz function $f$ over any bounded feature space $\mathcal{X}_1, \mathcal{X}_2$, while imposing the limitation that the input features are latent. This is applicable to a wide variety of problems, including recommendation systems, but also includes social network analysis, community detection, crowdsourcing, and product demand prediction. Many parametric models (e.g. low rank assumptions) can be framed as a specific case of our model.

Despite the generality and limited assumptions of our model, we present a simple similarity based estimator, and we provide theoretical guarantees bounding its error within the noise level $\gamma^2$. The analysis provides theoretical grounds for the popularity of similarity based methods. To the best of our knowledge, this is the first provable guarantee on the performance of neighbor-based collaborative filtering within a fully nonparametric model. Our algorithm and analysis follows from local Taylor approximation, along with an observation that the sample variance between rows or columns is a good indicator of "closeness", or the similarity of their function values. The algorithm essentially estimates the local metric information between the latent features from observed data, and then performs local smoothing in a similar manner as classical kernel regression.

Due to the local nature of our algorithm, our sample complexity does not depend on the latent dimension, whereas Chatterjee's USVT estimator [6] requires sampling almost every entry when the latent dimension is large. This difference is due to the fact that Chatterjee's result stems from showing that a Lipschitz function can be approximated by a piecewise constant function, which upper bound the rank of the target matrix. This discretization results in a large penalty with regards to the dimension of the latent space. Since our method follows from local approximations, we only require sufficent sampling such that locally there are enough close neighbor points.

The connection of our framework to regression implies many natural future directions. We can extend model (1) to multivariate functions $f$, which translates to the problem of higher order tensor completion. Variations of the algorithm and analysis that we provide for matrix completion can extend to tensor completion, due to the flexible and generic assumptions of our model. It would also be useful to extend the results to capture general noise models, sparser sampling regimes, or mixed models with both parametric and nonparametric or both latent and observed variables.

**Acknowledgements:**    This work is supported in parts by ARO under MURI award 133668-5079809, by NSF under grants CMMI-1462158 and CMMI-1634259, and additionally by a Samsung Scholarship, Siebel Scholarship, NSF Graduate Fellowship, and Claude E. Shannon Research Assistantship.

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
