[Supplementary Material · main_supplementary.pdf]

## A  Detailed Proof of Theorem 1

*Proof.* For any evaluation set of unobserved entries $E$, the expectation of $\varepsilon$-risk is

$$\mathbb{E}[\text{Risk}_\varepsilon] = \frac{1}{|E|} \sum_{(u,i) \in E} \mathbb{P}(|f(\mathbf{x}_1(u), \mathbf{x}_2(i)) - \hat{y}(u,i)| > \varepsilon) = \mathbb{P}(|f(\mathbf{x}_1(u), \mathbf{x}_2(i)) - \hat{y}(u,i)| > \varepsilon),$$

because the indexing of the entries are exchangeable and identically distributed. Therefore, in order to bound the expected risk, it is sufficient to provide a tail bound for the probability of the estimation error. For readability, we define the following events: with $\beta = np^2/2$,

- Let $A$ denote the event that $|\mathcal{S}_u^\beta(i)| \in [(m-1)p/2, 3(m-1)p/2]$.
- Let $B$ denote the event that $\min_{v \in \mathcal{S}_u^\beta(i)} \sigma^2_{\mathbf{x}_1(u)\mathbf{x}_1(v)} < \rho$.
- Let $C$ denote the event that $\left|\mu_{\mathbf{x}_1(u)\mathbf{x}_1(v)} - m_{uv}\right| < \alpha$ for all $v \in \mathcal{S}_u^\beta(i)$.
- Let $D$ denote the event that $\left|s_{uv}^2 - (\sigma^2_{\mathbf{x}_1(u)\mathbf{x}_1(v)} + 2\gamma^2)\right| < \rho$ for all $v \in \mathcal{S}_u^\beta(i)$.

Consider the following:

$$\mathbb{P}(\,|f(\mathbf{x}_1(u), \mathbf{x}_2(i)) - \hat{y}(u,i)| > \varepsilon\,)$$
$$\leq \mathbb{P}(\,|f(x_1(u), x_2(i)) - \hat{y}(u,i)| > \varepsilon\,|\,A, B, C, D) + \mathbb{P}(A^c) + \mathbb{P}(B^c|A) + \mathbb{P}(C^c|A,B) + \mathbb{P}(D^c|A,B,C). \tag{11}$$

Now,

$$\mathbb{P}\Big(A^c\Big) = \mathbb{P}\Big(\Big(|\mathcal{S}_u^\beta(i)| \notin \Big[\frac{(m-1)p}{2}, \frac{3(m-1)p}{2}\Big]\Big)\Big) \leq 2\exp\Big(-\frac{(m-1)p}{12}\Big) + (m-1)\exp\Big(-\frac{np^2}{8}\Big), \tag{12}$$

using Lemma 1. Similarly, using Lemma 2

$$\mathbb{P}(B^c|A) \leq \left(1 - h\left(\sqrt{\frac{\rho}{L^2}}\right)\right)^{\frac{(m-1)p}{2}} \leq \exp\left(-\frac{(m-1)p\,h\left(\sqrt{\frac{\rho}{L^2}}\right)}{2}\right). \tag{13}$$

Given choice of parameters, i.e. choice of $m$ and $p$ large enough for a given $\rho$, as we shall argue, the right hand side of (13) will be going to 0, and hence definitely less than $1/2$. That is, $\mathbb{P}(B|A) \geq 1/2$. Using this fact and Bayes formula, we have

$$\mathbb{P}(C^c|A,B) \leq 2\mathbb{P}(C^c|A) = 2\mathbb{P}\left(\cup_{v \in \mathcal{S}_u^\beta(i)}\left\{\left|\mu_{\mathbf{x}_1(u)\mathbf{x}_1(v)} - m_{uv}\right| > \alpha\right\}\,|\,A\right)$$
$$\leq 3(m-1)p\exp\left(-\frac{3np^2\alpha^2}{12B^2 + 4B\alpha}\right), \tag{14}$$

where last inequality follows from union bound, Lemmas 3 and choice of $\beta = np^2/2$. Again, choice of parameters, i.e. $m, n, p$ and $\alpha$ will be such that we will have the right hand side of (14) going to 0 and definitely less than $1/8$. Using this and arguments as used above based on Bayes' formula, we bound

$$\mathbb{P}(D^c|A,B,C) \leq \frac{\mathbb{P}(D^c|A)}{\mathbb{P}(B|A)\mathbb{P}(C|A,B)} \leq 4\mathbb{P}(D^c|A).$$
$$= 4\mathbb{P}\left(\cup_{v \in \mathcal{S}_u^\beta(i)}\left\{\left|s_{uv}^2 - (\sigma^2_{x_1(u)x_1(v)} + 2\gamma^2)\right| > \rho\right\}\,\Big|\,A\right)$$
$$\leq 12(m-1)p\exp\left(-\frac{\beta\rho^2}{4B^2(2LB_{\mathcal{X}}^2 + 4\gamma^2 + \rho)}\right) \tag{15}$$

where last inequality follows from union bound and Lemma 4.

Finally, with the choice of $\alpha = \beta^{-1/3}$, which is $\left(\frac{np^2}{2}\right)^{-1/3}$ since $\beta = \frac{np^2}{2}$, using Lemma 5, we obtain that

$$\mathbb{P}\left(|f(x_1(u), x_2(i)) - \hat{y}(u, i)| > \varepsilon \mid A, B, C, D\right) \leq \frac{3\rho + \gamma^2}{\varepsilon^2}\left(1 - \frac{\alpha}{\varepsilon}\right)^{-2}$$

$$\leq \frac{3\rho + \gamma^2}{\varepsilon^2}\left(1 + \frac{3\alpha}{\varepsilon}\right). \tag{16}$$

where we have used the fact that for given choice of $\alpha$ (since $\varepsilon$ is fixed), as $m$ increases, the term $\alpha/\varepsilon$ becomes less than $1/5$; for $x \leq 1/5$, $(1-x)^{-2} \leq (1+3x)$. If $p = \omega(m^{-1})$ and $p = \omega(n^{-1/2})$, all error terms from (12) to (15) diminish to 0 as $m, n \to \infty$. Specifically, if we choose $p = \max(m^{-1+\delta}, n^{-1/2+\delta})$, then putting everything together, we obtain (we assume that $m/2 \leq m - 1 \leq m$)

$$\mathbb{P}(|f(x_1(u), x_2(i)) - \hat{y}(u, i)| > \varepsilon)$$

$$\leq \frac{3\rho + \gamma^2}{\varepsilon^2}\left(1 + \frac{3\sqrt[3]{2}}{\varepsilon}n^{-\frac{2}{3}\delta}\right) + 2\exp\left(-\frac{1}{24}m^\delta\right) + m\exp\left(-\frac{1}{8}n^{2\delta}\right)$$

$$+ \exp\left(-\frac{1}{4}h\left(\sqrt{\frac{\rho}{L^2}}\right)m^\delta\right) + 3m^\delta\exp\left(-\frac{1}{5B^2}n^{\frac{2}{3}\delta}\right)$$

$$+ 12m^\delta\exp\left(-\frac{\rho^2}{8B^2(2LB_\mathcal{X}^2 + 4\gamma^2 + \rho)}n^{2\delta}\right).$$

The above bound holds for any $\rho > 0$, though as $\rho \to 0$, $m, n$ also need to increase accordingly such that $h\left(\sqrt{\frac{\rho}{L^2}}\right)$ is not too small, and $\rho$ must be $\omega(n^{-\delta})$ in order for the last term to vanish. We will impose that $\rho = \omega(n^{-2\delta/3})$ so that the last term is dominated by the second to last term. When the support of $P_\mathcal{X}$ is finite, then

$$h\left(\sqrt{\frac{\rho}{L^2}}\right) \geq \min_{x \in \mathcal{X}} P_\mathcal{X}(x),$$

such that the above bound holds even when $\rho = 0$. $\qquad\square$

# B   Useful Lemmas and their Proofs

This section presents key Lemmas that are utilized as part of the proof of Theorem 1. Lemma 1 establishes that as long as $p$ is large enough, then there are sufficiently large number of rows and columns that have overlap with row and column of a given candidate entry $(u, i)$. Lemma 2 establishes that there exists a row $v$ so that it's variance with respect to row $u$ is small. Lemmas 3 and 4 prove that the sample mean and sample variance of difference between a pair of rows are good proxy of the actual mean and variances. Collectively, these help establish in Lemma 5 that the true variance between $u$ and $u^*$, the row utilized by nearest neighbor user-user algorithm, is indeed small. These collection of results are established using known inequalities, namely Chernoff, Bernstein and Maurer-Pontil, stated in Section C for completeness.

## B.1   Sufficient overlap

Recall that $N_1(u)$ represents set of all column indices $j$ where $y(u, j)$ is observed. Similarly, $N_2(i)$ is the set of all row indices $v$ for which $y(v, i)$ is observed. For a pair of row indices $u, v$, $N_1(u, v) = N_1(u) \cap N_1(v)$. For a given $\beta \geq 2$, the set of all rows $v$ that can lead to a feasible estimation of $(u, i)$ as per the user-user nearest neighbor algorithm, denoted as $\mathcal{S}_u^\beta(i)$, is defined as

$$\mathcal{S}_u^\beta(i) = \left\{v \ : \ v \in N_2(i), |N_1(u, v)| \geq \beta\right\}.$$

Thus, establishing that $|\mathcal{S}_u^\beta(i)| \neq 0$ (better yet, $\gg 0$) leads to a guarantee that algorithm will be able to estimate missing entry at index $(u, i)$. The next Lemma provides sufficient condition for this event.

**Lemma 1.** *Given $p > 0$, $2 \leq \beta \leq np^2/2$ and $\alpha > 0$, for any $(u, i) \in [m] \times [n]$,*

$$\mathbb{P}\left(|\mathcal{S}_u^\beta(i)| \notin (1 \pm \alpha)(m-1)p\right) \leq 2\exp\left(-\frac{\alpha^2(m-1)p}{3}\right) + (m-1)\exp\left(-\frac{np^2}{8}\right).$$

*Proof.* The set $\mathcal{S}_u^\beta(i)$ consists of all rows $v$ such that (a) entry $(v, i)$ is observed, and (b) $|N_1(u, v)| \geq \beta$. For each $v$, define binary random variables $Q_v$ and $R_v$, where $Q_v = 1$ if $(v, i)$ is observed and 0 otherwise; $R_v = 1$ if $|N_1(u, v)| \geq \beta$ and 0 otherwise. Then, $|\mathcal{S}_u^\beta(i)| = \sum_{v \neq u} Q_v R_v$. Since $Q_v, R_v$ are binary variables and number of different $v \neq u$ are $m - 1$, we obtain that for any $0 \leq a < b \leq m - 1$,

$$\mathbb{P}\Big(|\mathcal{S}_u^\beta(i)| \notin [a, b]\Big) \leq \mathbb{P}\Big(\sum_{v \neq u} Q_v \notin [a, b]\Big) + \mathbb{P}\Big(\sum_{v \neq u} R_v < m - 1\Big). \tag{17}$$

Given that entries for each row $v$ are sampled independently, we have that $\sum_{v \neq u} Q_v$ is Binomial with parameters $(m - 1)$ and $p$. For choice of $a = (1 - \alpha)(m - 1)p$ and $b = (1 + \alpha)(m - 1)p$, a direct application of Chernoff's bound (see Section C for detail) implies that

$$\mathbb{P}\Big(\sum_{v \neq u} Q_v \notin [(1 - \alpha)(m - 1)p, (1 + \alpha)(m - 1)p]\Big) \leq 2 \exp\Big(-\frac{\alpha^2(m - 1)p}{3}\Big). \tag{18}$$

For $R_v = 1$, we require that $|N_1(u, v)| \geq \beta$. Given the sampling distribution, $N_1(u, v)|$ is Binomial with parameters $n$ and $p^2$. Therefore, for $\beta \leq np^2/2$, by another application of Chernoff's bound for lower tail, we obtain

$$\mathbb{P}\Big(R_v = 0\Big) \leq \exp\Big(-\frac{np^2}{8}\Big). \tag{19}$$

That is,

$$\mathbb{P}\Big(\sum_{v \neq u} R_v < m - 1\Big) \leq \sum_{v \neq u} \mathbb{P}\Big(R_v = 0\Big) \leq (m - 1) \exp\Big(-\frac{np^2}{8}\Big). \tag{20}$$

From (17)-(20), we obtain the desired result. $\qquad \square$

### B.2 Existence of a good neighbor

In order to show that a good-quality neighbor can be detected through sample variance, we need to show there exists a neighbor row whose true sample variance is small. Recall that latent space $\mathcal{X}_1$ is compact and bounded, $f$ is Lipschitz. We shall assume that the distribution $P_{\mathcal{X}_1}$ allows every nontrivial ball around any sample point in $\mathcal{X}_1$ obtained by sampling as per $P_{\mathcal{X}_1}$ have a positive measure. Under these conditions, next Lemma states that there exists a close neighbor for every point with high probability.

**Lemma 2.** *Let $(\mathcal{X}_1, P_{\mathcal{X}_1})$ admit a nondecreasing function $h : \mathbb{R}_{++} \to (0, 1]$ satisfying*

$$P_{\mathcal{X}_1}(\mathbf{x} \in B(x_0, r)) \geq h(r), \quad \forall x_0 \in \mathcal{X}_1, r > 0,$$

*where $B(x_0, r) \triangleq \{x \in \mathcal{X}_1 : d_{\mathcal{X}_1}(x, x_0) \leq r\}$. Consider $u \in [n]$ and set $\mathcal{S} \subset [n] \setminus \{u\}$. Then for any $\rho > 0$,*

$$\mathbb{P}\left(\min_{v \in \mathcal{S}} \sigma_{\mathbf{x}_1(u)\mathbf{x}_1(v)}^2 > \rho\right) \leq \left(1 - h\left(\sqrt{\frac{\rho}{L^2}}\right)\right)^{|\mathcal{S}|}.$$

*Proof.* Recall that $\sigma_{ab}^2 \triangleq \mathrm{Var}_{\mathbf{x} \sim P_{\mathcal{X}_2}}[f(a, \mathbf{x}) - f(b, \mathbf{x})]$, for any $a, b \in \mathcal{X}_1$. By Lipschitz property of $f$, we have that for any $x \in \mathcal{X}_2$,

$$|f(a, x) - f(b, x)| \leq L d_{\mathcal{X}_1}(a, b). \tag{21}$$

Therefore, it follows that

$$\sigma_{ab}^2 = \mathrm{Var}[f(a, \mathbf{x}) - f(b, \mathbf{x})] \leq \mathbb{E}[(f(a, \mathbf{x}) - f(b, \mathbf{x}))^2]$$
$$\leq L^2 d_{\mathcal{X}_1}(a, b)^2. \tag{22}$$

Now,

$$\mathbb{P}\left(\min_{v \in \mathcal{S}} \sigma_{\mathbf{x}_1(u)\mathbf{x}_1(v)}^2 > \rho\right) = \mathbb{P}(\cap_{v \in \mathcal{S}} \sigma_{\mathbf{x}_1(u)\mathbf{x}_1(v)}^2 > \rho) = \mathbb{P}\Big(\sigma_{\mathbf{x}_1(u)\mathbf{x}_1(v)}^2 > \rho\Big)^{|\mathcal{S}|},$$

where the last equality uses independence across sampling of $\mathbf{x}_1(v)$ for different $v$ and identical distribution, $P_{\mathcal{X}_1}$. From (22), it follows that if $\sigma^2_{\mathbf{x}_1(u)\mathbf{x}_1(v)} > \rho$ then $d_{\mathcal{X}_1}(\mathbf{x}_1(u), \mathbf{x}_1(v)) > \sqrt{\rho/L^2}$. Therefore, using definition of $h$, we obtain that

$$
\begin{aligned}
\mathbb{P}\Big(\sigma^2_{\mathbf{x}_1(u)\mathbf{x}_1(v)} > \rho\Big) &\leq \mathbb{P}\Big(d_{\mathcal{X}_1}(\mathbf{x}_1(u), \mathbf{x}_1(v)) > \sqrt{\frac{\rho}{L^2}}\Big) \\
&= \Big(1 - \mathbb{P}\Big(d_{\mathcal{X}_1}(\mathbf{x}_1(u), \mathbf{x}_1(v)) \leq \sqrt{\frac{\rho}{L^2}}\Big)\Big) \\
&\leq \Big(1 - h\Big(\sqrt{\frac{\rho}{L^2}}\Big)\Big).
\end{aligned}
$$

Putting all of the above together, we obtain the desired result. $\qquad\square$

**How does $h$ look like?** In order to provide some understanding toward the assumption on distribution $P_{\mathcal{X}_1}$, observe that the function $h(\cdot)$ is a form of the cumulative distribution function (CDF) for $P_{\mathcal{X}_1}$. The only distribution which does not satisfy this property is a distribution which has non-atomic isolated points. However, these isolated points have measure zero, such that they will never appear in our datasets with probability 1. We provide a few examples of distributions and their corresponding functions $h(\cdot)$.

**Example 1** (extremely uniform). *Suppose that $\mathcal{X} = \times_{k=1}^{d}[a_i, b_i] \in \mathbb{R}^d$ equipped with $L_\infty$ norm and $P_{\mathcal{X}}$ is a uniform distribution over $\mathcal{X}$. We can see that the function $h(r) := \prod_{k=1}^{d} \min\left\{1, \frac{r}{b_i - a_i}\right\}$ satisfies the condition $P_{\mathcal{X}}\left(\mathbf{x} \in B(x_0, r)\right) \geq h(r), \quad \forall x_0 \in \mathcal{X}, \forall r > 0.$*

**Example 2** (extremely clustered). *Suppose that $\mathcal{X} = \{x_1, \ldots, x_d\}$ equipped with the discrete topology and $P_{\mathcal{X}}$ is expressed in terms of its pmf $P_{\mathcal{X}}(x_k) = p_k$ with $\sum_{k=1}^{d} p_k = 1$. We can see that the function $h(r) := \min_k p_k$ works for $(\mathcal{X}, P_{\mathcal{X}})$.*

## B.3 Concentration of Sample Mean and Sample Variance

**Lemma 3.** *Given $u, v \in [m]$, $i \in [n]$ and $\beta \geq 2$, for any $\alpha > 0$,*

$$
\mathbb{P}\left(\left|\mu_{\mathbf{x}_1(u)\mathbf{x}_1(v)} - m_{uv}\right| > \alpha \mid v \in \mathcal{S}_u^\beta(i)\right) \leq \exp\left(-\frac{3\beta\alpha^2}{6B^2 + 2B\alpha}\right),
$$

*where recall that $B = 2(LB_{\mathcal{X}} + B_\eta)$.*

*Proof.* Given $\mathbf{x}_1(u) = x_1(u), \mathbf{x}_1(v) = x_1(v)$, the mean $\mu_{x_1(u)x_1(v)}$ is a constant. Recall that empirical mean $m_{uv}$ is defined as

$$
m_{uv} = \frac{1}{|N_1(u,v)|}\Big(\sum_{j \in N_1(u,v)} y(u,j) - y(v,j)\Big). \tag{23}
$$

The variable $\mathbf{x}_2(j)$ is sampled as per $P_{\mathcal{X}_2}$, independently from $x_1(u), x_1(v)$. And the noise term in each of the observation is independent zero-mean variable. Therefore, conditioned on $\mathbf{x}_1(u) = x_1(u), \mathbf{x}_1(v) = x_1(v)$, we have independent random variable, $Z(j) = y(u,j) - y(v,j)$ for $j \in N_1(u,v)$, that have mean $\mu_{x_1(u)x_1(v)}$. That is, $\tilde{Z}(j) = Z(j) - \mu_{x_1(u)x_1(v)}$, $j \in N_1(u,v)$ are zero-mean independent random variables. And by definition, each of them is bounded as

$$
|\tilde{Z}(j)| \leq 2B_\eta + LB_{\mathcal{X}} \leq 2(LB_{\mathcal{X}} + B_\eta) = B. \tag{24}
$$

In summary, conditioned on $\mathbf{x}_1(u) = x_1(u), \mathbf{x}_1(v) = x_1(v)$ and $N_1(u,v)$, $\mu_{x_1(u)x_1(v)} - m_{uv}$ is the average of $N_1(u,v)$ independent, zero mean random variables $\tilde{Z}(j)$, each of which have absolute value bounded above by $B$. Therefore, an application of Bernstein's inequality imply that

$$
\mathbb{P}\left(\left|\mu_{x_1(u)x_1(v)} - m_{uv}\right| > \alpha \mid \mathbf{x}_1(u) = x_1(u), \mathbf{x}_1(v) = x_1(v), N_1(u,v)\right) \leq \exp\left(-\frac{3|N_1(u,v)|\alpha^2}{6B^2 + 2B\alpha}\right).
$$

$$
\tag{25}
$$

When $v \in \mathcal{S}_u^\beta(i)$, $|N_1(u,v)| \geq \beta$. Further, since above holds for all possibilities of $x_1(u), x_2(v)$, we conclude that

$$\mathbb{P}\left(\left|\mu_{\mathbf{x}_1(u)\mathbf{x}_1(v)} - m_{uv}\right| > \alpha \,|\, v \in \mathcal{S}_u^\beta(i)\right) \leq \exp\left(-\frac{3\beta\alpha^2}{6B^2 + 2B\alpha}\right),$$

$\square$

Next we establish the concentration of the sample variance.

**Lemma 4.** *Given $u \in [m]$, $i \in [n]$, and $\beta \geq 2$, for any $\rho > 0$,*

$$\mathbb{P}\left(\left|s_{uv}^2 - (\sigma_{\mathbf{x}_1(u)\mathbf{x}_1(v)}^2 + 2\gamma^2)\right| > \rho \;\Big|\; v \in \mathcal{S}_u^\beta(i)\right) \leq 2\exp\left(-\frac{\beta\rho^2}{4B^2(2LB_{\mathcal{X}}^2 + 4\gamma^2 + \rho)}\right),$$

*where recall that $B = 2(LB_{\mathcal{X}} + B_\eta)$.*

*Proof.* Recall $\sigma_{ab}^2 \triangleq \mathrm{Var}[f(a, \mathbf{x}) - f(b, \mathbf{x})]$ for $a, b \in \mathcal{X}_1$, $\mathbf{x} \sim P_{\mathcal{X}_2}$, and sample variance between rows $u$ $v$ is defined as

$$s_{uv}^2 = \frac{1}{2|N_1(u,v)|(|N_1(u,v)| - 1)} \sum_{j,j' \in N_1(u,v)} ((y(u,j) - y(v,j)) - (y(u,j') - y(v,j')))^2$$

$$= \frac{1}{|N_1(u,v)| - 1} \sum_{j \in N_1(u,v)} (y(u,j) - y(v,j) - m_{uv})^2.$$

Conditioned on $\mathbf{x}_1(u) = x_1(u), \mathbf{x}_1(v) = x_1(v)$, we obtain that $\mathbb{E}[s_{uv}^2] = \sigma_{x_1(u)x_1(v)}^2 + 2\gamma^2$, with respect to randomness induced by $P_{\mathcal{X}_2}$ for sampling latent parameters for columns. Further, $X(j) = y(u,j) - y(v,j)$ are independent random variables conditioned on $\mathbf{x}_1(u) = x_1(u), \mathbf{x}_1(v) = x_1(v)$. Using the fact that $f$ is Lipschitz, space is bounded and noise is bounded, as before, we obtain that

$$|X(j)| = |y(u,j) - y(v,j)| \leq 2(LB_{\mathcal{X}} + B_\eta) = B.$$

Given this, by an application of Maurer-Pontil inequality (see Section C), we obtain that

$$\mathbb{P}\left(\left|s_{uv}^2 - (\sigma_{x_1(u)x_1(v)}^2 + 2\gamma^2)\right| > \rho \,|\, v \in \mathcal{S}_u^\beta(i), \mathbf{x}_1(u) = x_1(u), \mathbf{x}_1(v) = x_1(v)\right)$$

$$\leq 2\exp\left(-\frac{\beta\rho^2}{4B^2(2(\sigma_{x_1(u)x_1(v)}^2 + 2\gamma^2) + \rho)}\right), \tag{26}$$

where we used the property that $v \in \mathcal{S}_u^\beta(i)$ implies $|N_1(u,v)| \geq \beta$. Using the Lipschitz property of $f$ and boundedness of $\mathcal{X}_1$, we can bound $\sigma_{x_1(u)x_1(v)}^2 \leq L^2 B_{\mathcal{X}}^2$ as before. Therefore, the right hand side of (26) can be bounded as

$$\leq 2\exp\left(-\frac{\beta\rho^2}{4B^2(2L^2 B_{\mathcal{X}}^2 + 4\gamma^2 + \rho)}\right). \tag{27}$$

Given that this bound is indepedent of $x_1(u), x_1(v)$, we can conclude the desired result. $\square$

### B.4 Concentration of Estimate

Now we establish the final step in the proof of Theorem 1. As in the proof of Theorem 1, for a given $(u, i)$ with $u \in [m]$, $i \in [n]$ and $\beta \geq 2$, define events

- Let $A$ denote the event that $|\mathcal{S}_u^\beta(i)| \in [(m-1)p/2, 3(m-1)p/2]$,
- Let $B$ denote the event that $\min_{v \in \mathcal{S}_u^\beta(i)} \sigma_{\mathbf{x}_1(u)\mathbf{x}_1(v)}^2 < \rho$,
- Let $C$ denote the event that $\left|\mu_{\mathbf{x}_1(u)\mathbf{x}_1(v)} - m_{uv}\right| < \alpha$ for all $v \in \mathcal{S}_u^\beta(i)$,
- Let $D$ denote the event that $\left|s_{uv}^2 - (\sigma_{\mathbf{x}_1(u)\mathbf{x}_1(v)}^2 + 2\gamma^2)\right| < \rho$ for all $v \in \mathcal{S}_u^\beta(i)$.

**Lemma 5.** *Under the setting described above and given $\alpha > 0$, $\rho > 0$ and $\varepsilon > \alpha$, under the algorithm user-user nearest neighbor, we have*

$$\mathbb{P}\left(\,|f(\mathbf{x}_1(u), \mathbf{x}_2(i)) - \hat{y}(u, i)| > \varepsilon \,|\, A, B, C, D\right) \leq \frac{3\rho + \gamma^2}{(\varepsilon - \alpha)^2}.$$

*Proof.* Under the algorithm user-user nearest neighbor, the error of the estimate is given by

$$f(\mathbf{x}_1(u), \mathbf{x}_2(i)) - \hat{y}(u, i) = f(\mathbf{x}_1(u), \mathbf{x}_2(i)) - y(u^*, i) - m_{uu^*}$$
$$= f(\mathbf{x}_1(u), \mathbf{x}_2(i)) - f(\mathbf{x}_1(u^*), \mathbf{x}_2(i)) - \eta_{u^*, i} - m_{uu^*}.$$

Given $\mathbf{x}_1(u) = x_1(u)$, $\mathbf{x}_1(u^*) = x_1(u^*)$ such that events $A, B, C$ and $D$ are satisfied, we have that

$$\mathbb{E}[f(x_1(u), \mathbf{x}_2(i)) - f(x_1(u^*), \mathbf{x}_2(i)) - \eta_{u^*, i}] = \mu_{x_1(u)x_1(u^*)}, \tag{28}$$

with respect to $\mathbf{x}_2(i) \sim P_{\mathcal{X}_2}$.

Conditioned on event $C$, that is, $\left|\mu_{x_1(u)x_1(v)} - m_{uv}\right| < \alpha$ for all $v \in \mathcal{S}_u^\beta(i)$, included $u^*$, it is sufficient to bound the probability of event

$$E = \left\{|f(x_1(u), \mathbf{x}_2(i)) - f(x_1(u^*), \mathbf{x}_2(i)) - \eta_{u^*, i} - \mu_{uu^*}| > \varepsilon - \alpha\right\}. \tag{29}$$

Conditioned on $\mathbf{x}_1(u) = x_1(u)$, $\mathbf{x}_1(u^*) = x_1(u^*)$,

$$\mathrm{Var}[f(x_1(u), \mathbf{x}_2(i)) - f(x_1(u^*), \mathbf{x}_2(i)) - \eta_{u^*, i}] = \sigma^2_{x_1(u)x_1(u^*)} + \gamma^2, \tag{30}$$

Therefore, by standard Chebychev's inequality, we obtain

$$\mathbb{P}\left(|f(x_1(u), \mathbf{x}_2(i)) - f(x_1(u^*), \mathbf{x}_2(i)) - \eta_{u^*, i} - \mu_{x_1(u)x_1(u^*)}| > \varepsilon - \alpha\right) \leq \frac{\sigma^2_{x_1(u)x_1(u^*)} + \gamma^2}{(\varepsilon - \alpha)^2}. \tag{31}$$

The selection of $u^*$ was done using empirical estimates $s^2_{uv}$ across $v \in \mathcal{S}_u^\beta(i)$. By condition on event $D$ happening, we have that for any $v \in \mathcal{S}_u^\beta(i)$, $s^2_{uv}$ is within $\rho$ of $(\sigma^2_{\mathbf{x}_1(u)\mathbf{x}_1(v)} + 2\gamma^2)$. And condition on event $B$, we have that there is at least one $v \in \mathcal{S}_u^\beta(i)$ so that $\sigma^2_{\mathbf{x}_1(u)\mathbf{x}_1(v)} < \rho$; let one such $v$ be denoted as $v^*$. Therefore, we obtain that

$$\sigma^2_{x_1(u)x_1(u*)} + 2\gamma^2 - \rho \leq s^2_{uu^*}$$
$$\leq s^2_{uv}$$
$$\leq \sigma^2_{x_1(u)\mathbf{x}_1(v)} + 2\gamma^2 + \rho$$
$$\leq 2\gamma^2 + 2\rho. \tag{32}$$

From above, we can conclude that $\sigma^2_{x_1(u)x_1(u^*)} \leq 3\rho$. Replacing this in (31), we obtain the bound on right hand side as

$$\leq \frac{3\rho + \gamma^2}{(\varepsilon - \alpha)^2}. \tag{33}$$

Since this bound holds for all choices of $\mathbf{x}_1(u), \mathbf{x}_1(u^*)$ conditioned on events $A, B, C$ and $D$, we conclude the desired result. $\qquad \square$

## C  Useful Inequalities

**Lemma 6** (Bernstein's Inequality). *If $X_1, \ldots X_n$ are independent zero-mean r.v. such that $|X_i| \leq M$ almost surely, then for all $t$,*

$$\mathbb{P}\left(\frac{1}{n}\sum_{i=1}^n X_i > t\right) \leq \exp\left(-\frac{3n^2t^2}{2(3\sum_j \mathbb{E}[X_j^2] + Mnt)}\right)$$
$$\leq \exp\left(-\frac{3nt^2}{6M^2 + 2Mt}\right).$$

**Lemma 7** (Chernoff's Inequality). *If $X_1, \ldots X_n$ are independent r.v. such that $X_i \in (0,1)$, and let $X$ denote their sum. Then for any $\delta \in (0,1)$,*

$$\mathbb{P}\left(X \leq (1-\delta)\mathbb{E}[X]\right) \leq \exp(-\delta^2 \mu/2),$$

*and for any $\delta > 0$,*

$$\mathbb{P}\left(X \geq (1+\delta)\mathbb{E}[X]\right) \leq \exp(-\delta^2 \mu/3)$$

**Lemma 8** (Maurer-Pontil Inequality [19]). *For $n \geq 2$, let $X_1, \ldots X_n$ be independent random variables such that $X_i \in (0,1)$. Let $V(X)$ denote their sample variance, i.e., $V(X) = \frac{1}{2n(n-1)} \sum_{i,j}(X_i - X_j)^2$. Let $\sigma^2 = \mathbb{E}[V(X)]$ denote the true variance. For any $\delta \in (0,1)$,*

$$\mathbb{P}\left(V(X) - \sigma^2 < -s)\right) \leq \exp\left(-\frac{(n-1)s^2}{2\sigma^2}\right),$$

*and*

$$\mathbb{P}\left(V(X) - \sigma^2 > s)\right) \leq \exp\left(-\frac{(n-1)s^2}{2\sigma^2 + s}\right).$$

The Maurer-Pontil Inequality implies the following corollary for all bounded random variables.

**Corollary 1.** *For $n \geq 2$, let $X_1, \ldots X_n$ be independent random variables such that $X_i \in (a,b)$. Let $V(X)$ denote their sample variance, i.e., $V(X) = \frac{1}{2n(n-1)} \sum_{i,j}(X_i - X_j)^2$. Let $\sigma^2 = \mathbb{E}[V(X)]$ denote the true variance. For any $\delta \in (0,1)$,*

$$\mathbb{P}\left(\left|V(X) - \sigma^2\right| < s)\right) \leq 2\exp\left(-\frac{(n-1)s^2}{(b-a)^2(2\sigma^2 + s)}\right).$$