[Reviews · NeurIPS 2016]

Reviewer 1

Summary

This paper proposes to frame the problem of collaborative filtering as one of nonparametric bi-linear regression. The authors propose an interesting technique for deriving a neighbourhood model that estimates user-item ratings. They show that the error of the estimator is bounded by a function of the distance between neighbours which guides their choice of neighbour weights. They also prove the consistency (as the number of users and items tend to infinity the true ratings are recovered) of their proposed estimator. The authors extend their approach to tensor factorization and empirically compare their approach to several well-known collaborative filtering baselines.

Qualitative Assessment

I found the paper to be well-written and relatively easy to follow. The proposed model, the algorithm, the consistency theorems and the study are interesting and appear sound. It would be worthwhile for the authors to discuss how widely applicable the task of blind regression is and how general are some of the techniques they have used (e.g., the Taylor expansion as a way to derive a useful "neighbourhood" estimator). Other tasks that involved modelling discrete data share commonalities with collaborative filtering (e.g., modelling collections of documents in which users and items are mapped to documents and words). Do the techniques suggested here naturally extend to these? When flattening a t-order tensor into a 2-order tensor it seems like you are effectively hiding some of the interesting structure in the data (e.g., if the 3rd-order is time you are masking users' change in preferences and changing item popularity). I am wondering how this may affect Theorem 2? Other comments: - There are a few things in the empirical study that I would suggest clarifying: a) Do you use a validation set to set the lambda hyper-parameter? b) It would be good to report the variance in addition to the reported means in Figures 1 &2. c) From a practical perspective it would be interesting to study how the performance of your method varies wrt the value of beta.

Confidence in this Review

1-Less confident (might not have understood significant parts)


Reviewer 2

Summary

This paper's main contributions are the following: (1) Simple nonparametric algorithms, with theoretical guarantees on the sample complexity, for collaborative filtering (CF) for matrix and tensor data. (2) Showing the connection of the proposed algorithms to the traditional CF methods such as user-user and item-item based CF, and consequently a justification of the effectiveness of such methods. The basic algorithms are simple and intuitive: under certain smoothness assumptions, the rating y_ui of a user (u) on an item (i) can be approximated as y_u'i + y_ui' - y_u'i' where u' denotes another user and i' denotes another item. Based on this idea, a new nonparametric algorithm is proposed which uses a weighted aggregate of this approximation considering a certain neighborhood of u and i (and in the end it looks somewhat akin to a weighted nearest neighbors like algorithm). The resulting algorithm is shown to be consistent as the number of users and item go to infinity. On a small set of experiments, the proposed algorithms are shown to be competition to conventional CF algorithms.

Qualitative Assessment

I think this is a nice paper that, in addition to presenting a simple method for doing collaborative filtering, tries to carefully explicate why the traditional collaborative filtering algorithms work well despite their simplicity. The algorithms are simple to implement and come with theoretical guarantees on the sample complexity and finite sample error bounds on the prediction. Some additional comments: - While the paper covers many of the relevant papers, I would also like to point out the following papers as well, that have looked at analyzing CF algorithms, e.g., "Using mixture models for collaborative filtering". Jon Kleinberg and Mark Sandler (2004); and "Convergent algorithms for collaborative filtering". Jon Kleinberg and Mark Sandler (2003) - Given the theoretical nature of the paper, I wasn't really expecting a lot of experiments but I would be keen to see how the proposed algorithms for CF compare against the other dominant approach for CF, i.e., low-rank matrix/tensor factorization Minor comment: - Text around line 69 seems incomplete (abruptly line break). - References [23] and [24] are the same.

Confidence in this Review

2-Confident (read it all; understood it all reasonably well)


Reviewer 3

Summary

This paper studied the problem of using nonparametric regression techniques for matrix and tensor completion tasks. The authors proposed a simple algorithm for such a purpose, and provided a theoretical result on the performance of their algorithm. The proposed method is also compared against various other nonparametric methods.

Qualitative Assessment

Overall I feel that the paper studies an interesting problem from a different angle (most of the recommender systems work these days are matrix-factorization-type approaches). The proposed blind regression approach is well motivated and theoretically justified, and the technical side seems to have enough quality. My main concern is the justification of the proposed approach. Such a nonparametric approach seems simple to implement, so that can be its strong suit (over more computationally intensive matrix-factorization algorithms). But there are no experimental results to support this. In terms of prediction accuracy, the proposed approach does outperform other simple baselines but do not beat matrix-factorization algorithms. This is not a good enough experimental justification of the proposed approach.

Confidence in this Review

1-Less confident (might not have understood significant parts)


Reviewer 4

Summary

The paper proposes new algorithms for matrix completion, which can be used in recommender systems. Whereas many approaches to matrix completion make some kind of low-rank assumption about the matrices in question, this paper is able to make certain recovery guarantees so long as the a particular Lipschitz condition is satisfied. This potentially makes the algorithm more flexible than existing approaches in terms of capturing real data.

Qualitative Assessment

The authors do a good job of presenting the high-level ideas behind their contribution and presenting the relevant literature in context. The actual contribution is quite technical in nature, but a good amount of effort is taken to walk the reader through it. The authors might also look into approaches like SLIM (Ning and Karypis), which also approach matrix completion tasks using (fairly simple) models that overcome the low-rank assumption of typical matrix completion approaches. Although the paper promises to recover matrix data generated by a quite general class of functions, I struggled to understand which of the operating assumptions (section 2) are actually realistic. In particular, assumption (e) (each entry is observed independently) is certainly violated in the netflix and movielens datasets where the "missing at random" assumption does not hold (as would be the case in any dataset where users self-select what to evaluate; see papers on the "missing not at random" assumption). But I don't know how big an issue this violation is in practice. The independence assumption between latent features of users and movies is also questionable (and stands in stark contrast to work on, say, social regularization in recommender systems, which critically depends on this assumption *not* being true). Certainly the paper considers a very interesting, general, and seemingly novel form of matrix completion, but the contribution comes into question a little bit if it depends on assumptions which definitely *do not* hold in the datasets being studied. This may be partly an issue that can be addressed by presentation -- i.e., the idealized algorithm depends on assumptions which may not be true, but how badly can these be violated before the algorithm stops working in practice? Obviously it works okay in practice, so maybe the work could be presented in a way that focuses less on the theoretical guarantees under these (probably too strict) assumptions, and rather focuses more on its practical use. Experiments on Netflix and movielens data are interesting, though rather on the proof-of-concept side, as they operate on small samples of the data rather than the subsets that are regularly used for evaluation. Comparisons are also against classical methods, rather than strong baselines for these datasets (e.g. many of the suite of standard algorithms available in MyMediaLite would presumably outperform the simple baselines considered here). That being said I acknowledge that outperforming the state-of-the-art (much of which is heavily tuned to these particular datasets) is not the point of these experiments, and the authors' claims are reasonably down to earth in this matter. The generalization to tensor factorization, and the results on inpainting data are interesting, but honestly a little bit of a distraction in a short-form paper, as this application is so different from what's covered during the rest of the paper. Since the method ultimately collapses the tensor to a matrix, it's a fairly straightforward extension that could be mentioned in passing and not given so much attention. It would be much more compelling to end the paper with a proper discussion and conclusion as opposed to these very brief inpainting experiments. Ultimately, I gave this paper a positive evaluation in most aspects: it's technically strong, it's novel enough, and the presentation is good. It's only practical aspects where I gave it a low rating, but this is a *big* issue given the practical applications that are being targeted! The strong model assumptions don't seem satisfiable in practice, and this is a hard issue to overlook.

Confidence in this Review

2-Confident (read it all; understood it all reasonably well)


Reviewer 5

Summary

This paper motivates a method for blind regression, using a nonparametric model for collaborative filtering within the domain of matrix completion. Using some weak assumptions of regularity and smoothness, the paper proposes a model whereby observed entries of the matrix are noisy observations of the output of an unknown Lipschitz function on features relating to a user and an item. In contrast to previous work on collaborative filtering, this paper considers the case where the features on the users and items are themselves latent. Using a multiple Taylor expansions, the paper presents an algorithm and provides a finite same error bound. Intuitively, the bound stems from the accuracy of local Taylor approximations under small amounts of noise and then makes use of the variance between two rows or columns of the unknown matrix as a bound on the squared error. This variance is estimated empirically on the known points. After the discussion of the algorithm and its error bound, the paper extends this matrix completion method to tensor completion in a setting with equivalent assumptions for tensors. This is achieved by partitioning the tensor into two disjoint subsets and flattening in the tensor in each of those subsets to create the rows and columns of a new matrix which satisfies all of the assumptions needed for their earlier blind regression algorithm for matrices. The paper closes by providing some experiments performed on the MovieLens and Netflix datasets, where the algorithm is shown to give competitive results compared to other collaborative filtering and matrix completion methods.

Qualitative Assessment

Explanation of ratings: Technical Aspect: I thought this paper did a good job of motivating a different approach to user-user or item-item collaborative filtering. It is impressive that equivalent performance (empirically) is achievable with far fewer assumptions on the data, and this is definitely something for practitioners to consider in their research. While less is assumed in the model, less is explained however. A more detailed analysis of the algorithm may benefit this paper. Novelty: This paper takes a very different, and non-restrictive approach to collaborative filtering and shows good results with it. By using successive Taylor approximations on their latent function, however, it seemed like the paper spent time motivating the idea of a latent function and latent variables but then assumed low noise and removed it from the model. In other words, all of the latency of this model appears to get wrapped into local approximations and then the model appears to proceed by working directly on the observed entries and ignoring the latent features and function entirely. I do think that this idea of weighting based on variance is novel as are the extensions to tensor completion and I would like to see more discussion of the later. Potential Impact: This paper has the potential for impact by motivating the idea of completing a matrix using the variance as a weighting scheme for selecting representative users and columns. Future theoretical work may well stem from this idea. Clarity: The paper was mostly clear. I appreciated the section intuitively motivating the algorithm. The section motivating the extension to tensors could be clearer, particularly in the decomposition of the tensor into disjoint sets. It is unclear to me how a matrix is formed from that and how this maps back to the original tensor after the matrix is “flattened.” Furthermore, the algorithm is not described as clearly as it could have been. I think that the steps of the algorithm are somewhat lost in their definition. Suggestions for revision: Clarify your algorithm section. It is not immediately obvious what you are trying to learn, and how you go about doing so. I understand the goal is to make predictions on unknown entries, but it is not clear where the actual regression is happening. For instance, in the user-user or item-item case, is the regression simply a search problem to find the corresponding rows and columns with minimum variance and weight them accordingly? Clarify where blind regression plays a role in this paper. It seems as though you motivate the concept of blind regression, but then show that it reduces to a weighting scheme on the variance of the difference between rows and columns. If there is room, a brief mention of the intuition behind your theorem would be useful. This paper begins to address this in the last paragraph of section 5, but given the length of the full proof, some added intuition would be useful here. In your abstract, you mentioned the algorithm shedding light on the success of collaborative filtering in practice. I am not sure that this question was fully answered. Perhaps you could add a couple of sentences in the intuition or contribution sections to more fully explain how your nearest-neighbor approach generalizes and explains the success of other, more restrictive models collaborative filtering.

Confidence in this Review

2-Confident (read it all; understood it all reasonably well)


Reviewer 6

Summary

This paper proposes a framework for performing nonparametric regression over latent variable models, motivated by matrix completion for recommendation systems. This framework is also elegantly extended to completing higher order tensors. An experimental comparison to competing methods for matrix and tensor completion shows that the proposed approach provides competitive or sometimes better predictive performance than these existing methods.

Qualitative Assessment

The framework for blind regression based on the first order local Taylor expansion of a function is interesting, and the connection to traditional item-item (or user-user) collaborative filtering methods is a useful and somewhat unexpected result. The presented extension to tensor completion has an elegant simplicity. The explanation of the proposed algorithm and the intuition behind it is well written. The discussion of experimental results could be improved somewhat. The paper suggests that the difference in performance between the proposed approach and competing matrix completion methods could due to the different underlying assumptions of the various approaches. This point and its connection to the true structure of the dataset should be explained in more detail. In Section 7 (Matrix Completion Experiments), it is stated that setting \beta = 2 ensures that the algorithm is able to compute an estimate for all missing entries. Why is this the case? This point should be explained/clarified. The results of the tensor completion experiments show that the proposed algorithm outperform the competing FBCP and TenAlt approaches in many cases. It would be helpful to explain why the proposed algorithm outperforms these approaches. The paper ends abruptly without a Conclusion section. A short conclusion section should be written; the discussion of related literature and/or contributions in the Introduction section could be shortened to make room for a conclusion section.

Confidence in this Review

2-Confident (read it all; understood it all reasonably well)